# Comparing variable and feature selection strategies for prediction - protocol of a simulation study in low-dimensional transplantation data

Linard Hoessly[1]*, Jaromil Frossard[1], Simon Schwab[2], Frédérique Chammartin[3], Alexander Leichtle[4], Peter Werner Schreiber[5], Dionysios Neofytos[6], Michael Koller[1], with the Swiss Transplant Cohort Study (STCS)[1]¶

1 Data Center of the Swiss Transplant Cohort Study, University Hospital Basel, Basel, Switzerland, 2 Swisstransplant, Bern, Switzerland, 3 Department of Clinical Research, University Hospital Basel, Basel, Switzerland, 4 Cantonal Hospital Baden, Baden, Switzerland, 5 Department of Infectious Diseases and Hospital Epidemiology, University Hospital Zurich and University Zurich, Zurich, Switzerland, 6 Transplant Infectious Diseases Unit, Service of Infectious Diseases, University Hospitals Geneva, University of Geneva, Geneva, Switzerland

¶ Membership list of the STCS can be found in the Acknowledgments.

* linarddavid.hoessly@usb.ch

**Data availability statement:** No datasets were generated or analysed during the current study.

## Abstract

The integration of machine learning methodologies has become prevalent in the development of clinical prediction models, often suggesting superior performance compared to traditional statistical techniques. Within the scope of low-dimensional datasets, encompassing both classical and machine learning paradigms, we plan to undertake a comparison of variable selection methodologies through simulation-based analysis. The principal aim is the comparison of the variable selection strategies with respect to relative predictive accuracy and its variability, with a secondary aim the comparison of descriptive accuracy. We use six distinct statistical learning approaches across both data generation and model learning. The present manuscript is a protocol for the corresponding simulation study registration (Study registration Open Science Framework ID: k6c8f). We describe the planned steps through the Aims, Data, Estimands, Methods, and Performance framework for simulation study design and reporting.

## 1 Introduction

Accurate probability predictions are vital for diagnostic and prognostic models, enabling effective risk assessment, mitigation strategies, and informed clinical decisions [1]. Both statistical regression and machine learning (ML) approaches are commonly used [1,2].

Variable selection significantly impacts the statistical quality and practical utility of prediction models [3,4]. In regression, it is termed variable selection, while in ML, it is known as feature selection [4]. Both concepts relate to variable importance, introduced by Breiman in random forests [5], which quantifies a variable's contribution to a model's predictive performance.

**Funding:** This project (FUP 214) has been facilitated by the Swiss Transplant Cohort Study. The Swiss Transplant Cohort Study is supported by the Swiss National Science Foundation (SNSF grant 33CS30_201385, https://www.snf.ch), Unimedsuisse and the Transplant Centers.

**Competing interests:** The authors have declared that no competing interests exist.

Breiman's distinction between the "data modeling" and "algorithmic modeling" cultures [6] highlights two statistical modeling approaches. The data modeling culture assumes an underlying probabilistic model to understand the data-generating process, whereas the algorithmic modeling culture prioritises predictive accuracy. ML models usually need large datasets with high-dimensional data to work well [19,25]. In contrast, clinical data are often in the setting of low-dimensional low sample size data, like those in transplantation medicine [15].

Despite limited non-asymptotic theory [7], data-dependent selection methods are widely used in practice [1,2,8]. Recent advancements in artificial intelligence have increased interest in applying ML techniques to predict patient outcomes, raising questions about the comparative performance of variable and feature selection across regression and ML models, which motivate our work.

For convenience, we use statistical learning and ML methods interchangeably, distinguishing where necessary. Statistical learning methods refer to parametric models, such as unpenalized logistic regression, while ML methods include algorithms like boosted trees, random forests, and multivariate adaptive regression splines (MARS) [9]. LASSO and Ridge regression can be classified as both. Similarly, we use variable selection for methods like backward p-value selection and feature selection for techniques like the Boruta algorithm, which selects variables using random forests. While classical variable selection methods have been well-studied in the data modeling culture [7], they are also employed in ML models [9]. In particular, we note that despite their shared goal [4,10], statistical and ML-based selection methods are often perceived as distinct due to differing terminology [11].

Various statistical learning and ML models are used in practice. The key models we consider are briefly introduced below.

Logistic regression is a widely used statistical model in medicine for understanding or predicting the effect of multiple risk factors (i.e., covariates) on a binary outcome [1]. Often, variable selection is performed before constructing the final model [1].

Random forests, boosted trees, MARS, LASSO, and Ridge are all ML techniques used for classification and regression [9]. Although some models inherently perform feature selection, pre-selection is often beneficial, as noted for random forests [12], boosted trees [13], or to improve computational efficiency and model interpretability [9].

The choice of an appropriate variable selection method depends heavily on the study's objectives [14], with limited guidance on what to avoid [15] and no clear consensus on best practices [10]. Statistical methods include significance-based approaches (e.g., univariable p-value selection), information criteria-based approaches (e.g., AIC), penalized likelihood methods (e.g., LASSO), and others, such as the change-in-estimate criterion [8]. For a review of statistical variable selection, see [7,8]. A detailed discussion of data-dependent selection methods and statistical challenges can be found in [10].

In supervised ML, feature selection techniques are commonly categorized as filters, wrappers, or embedded methods [4]. Filters preselect predictors independently of the learning algorithm, while wrappers alternate between selection and modeling. For instance, univariable p-value-based selection is a filter, whereas backward p-value-based selection in logistic regression is a wrapper [4]. Embedded methods integrate selection into model-building, with LASSO being a well-known example. For further discussion on feature selection methods, see [16].

The data used in this study is based on the Swiss Transplant Cohort Study (STCS), a prospective, multicenter study enrolling all solid organ transplant (SOT) recipients in Switzerland since May 2008 [17]. The STCS standardizes data collection across transplant centers, aiding clinical research, quality control, and outcome monitoring. Organ transplantation is

a complex, multidisciplinary field requiring tailored statistical methods. However, due to the modest number of available organs and patients, coupled with the increasing volume of patient data, researchers must carefully select variables for prediction models [17].

The estimation of the appropriate number of variables is relatively restrictive for binary and time-to-event models, often following the one-in-ten rule [2], which recommends at most one variable per 10 events, or using more advanced methods [18]. In contrast, ML methods typically require more observations than classical models, raising concerns about their applicability in limited-data settings like transplantation medicine [19].

In real-world data analysis, the true underlying distributions of variables and outcomes are often unknown, making simulation studies an essential evaluation tool [20]. Simulated data have known distributions, allowing researchers to explore different scenarios and study method properties. While theoretical statistics often focus on asymptotic results, finite sample properties in real data remain uncertain, underscoring the importance of simulations for understanding variable and feature selection methods [7,10]. We follow the Aims, Data, Estimands, Methods, and Performance (ADEMP) framework [20] for structured planning and reporting of this simulation study.

While simulation studies are often used to highlight the advantages of new methods [21], our goal is to maintain objectivity and conduct a balanced comparison with transparent reporting [22]. To ensure fairness, we will involve experts familiar with the methods, without prior biases or conflicts of interest.

Our simulations will generate outcomes using six different data-generating processes and subsample predictors from a real STCS dataset with simulated outcomes. The model-building process consists of a variable selection step followed by model fitting, with the remaining complement set of the subsample serving as the evaluation set.

This study protocol aims to systematically investigate and compare the following:

1. Statistical variable selection methods,
2. Feature selection techniques from ML,
3. The performance of these selection methods in conjunction with different models, including logistic regression and ML-based approaches.

We will compare the following variable/feature selection strategies: univariate threshold-based selection, k-best selection, and backward selection. These will be evaluated using various scores, including p-value, AIC, CAR-score, permutation importance, information gain, minimal joint mutual information maximization, impurity importance, and symmetric uncertainty. Additionally, we will consider the random-forest-based BORUTA algorithm [23] and LASSO [24].

The data-generating process strategy builds on previous research, particularly studies comparing logistic regression to ML models [25] and work exploring the 'data-hungriness' of various statistical and ML models [19].

This study primarily compares variable and feature selection methods across different ML and statistical learning models, focusing on predictive performance. A secondary aim is to investigate descriptive performance, aligning with key statistical challenges identified in [10]. We will sample predictors from real data and use an estimated data-generating mechanism (DGM) that allows for nonlinearity in continuous predictors to better reflect clinical data.

Other studies have examined the impact of feature or variable selection on predictive performance. [26] compared regression-based and tree-based methods, finding that tree-based selection led to higher parsimony in large datasets, while regression-based methods favored

smaller datasets. [27] investigated 22 feature selection methods across different models, evaluating accuracy and computation time. More recently, [28] reviewed and analyzed test-based, penalty-based, and screening-based variable selection methods for logistic regression, while both [29] and [30] plan to also consider predictive performance measures as a secondary outcome for variable selection before logistic regression or linear regression.

In Sect 2 we give an introduction to the simulation design, and detail the description of each part of the ADEMP structure. In particular, Sect 2.2 describes the aims of the simulation study, Sect 2.3 describes the planned data-generation mechanism, Sect 2.4 the target estimands. Then Sect 2.5 briefly describes the variable selection methods and regression/ML models, while we describe the performance measures in Sect 2.6.

## 1.1 Acknowledgments

We thank Sandor Balog for a helpful exchange on feature selection in machine learning, and Korbinian Strimmer for a helpful exchange on the CAR score. We thank Georg Heinze, Willi Sauerbrei and Edwin Kipruto for helpful discussions on variable selection.

# 2 Simulation design trough the ADEMP structure

## 2.1 Overview table

See Tables 1 and 2.

**Table 1. Overview over ADE (from ADEMP) table.**

| Aims: | • To compare variable and feature selection methods concerning their performance for prediction.<br>• To compare the variable and feature selection methods concerning their descriptive performance. |
|---|---|
| Data generating mechanism: | →**Training/development dataset:**<br>**Sample distribution for predictors:**<br>• The predictors are sampled from the real population.<br>**True models:**<br>The models are estimated based on all data, where four variables are not included in the true models by construction (for more see SI Sect 3).<br>• DGM1: Unpenalised logistic regression with continuous predictors using restricted cubic smoothing splines.<br>• DGM2: Penalised logistic regression via LASSO (all continuous predictors linear).<br>• DGM3: Penalised logistic regression via RIDGE (all continuous predictors linear).<br>• DGM4: Random forest.<br>• DGM5: Boosted trees.<br>• DGM6: Multivariate adaptive regression splines.<br>**Sample size:**<br>Sample sizes considered are the following $n = \{250, 500, 1000\}$.<br>**Number of DGM scenarios and simulation runs:**<br>$\|M\| \cdot \|n\| = 6 \cdot 3 = 18$ scenarios<br>$N = 1500$ simulation repetitions per scenario<br>**Evaluation dataset:**<br>The evaluation dataset is then the complement to the predictors sampled from the real population, with sampled outcome. |
| Estimand/Target of analysis: | • Model prediction error<br>• Model discrimination, sharpness, and calibration<br>• Inclusion of true and false predictors |

**Table 2. Overview over MP (from ADEMP) table.**

| Methods: | Variable selection strategies (age and sex kept in the model): | |
|---|---|---|
| | • **Backward variable/feature selection strategies** | |
| | Methods | Parameters |
| | Logistic regression p-value based | 0.1, 0.2, 0.5 |
| | Logistic regression AIC based | - |
| | | |
| | • **Threshold-based univariate selection strategies** | |
| | Method | Parameters |
| | Logistic regression p-value based | 0.1, 0.2, 0.5 |
| | | |
| | • ***k*-best univariate selection strategies** | |
| | Method | Scores |
| | 7-best | Logistic regression p-value based<br>CAR<br>Permutation importance<br>Information gain<br>Minimal joint mutual information maximization<br>Impurity importance<br>Symmetric uncertainty |
| | 14-best | Logistic regression p-value based<br>CAR<br>Permutation importance<br>Information gain<br>Minimal joint mutual information maximization<br>Impurity importance<br>Symmetric uncertainty |
| | | |
| | • **Other strategies** | |
| | Method | Parameters |
| | LASSO | - |
| | Boruta | - |
| | | |
| | **Regression/ML models:**<br>The following models are estimated based on the simulated data.<br>• R1: Unpenalised logistic regression with continuous predictors<br>using restricted cubic smoothing splines.<br>• R2: Penalised logistic regression via LASSO (all continuous predictors linear).<br>• R3: Penalised logistic regression via RIDGE (all continuous predictors linear).<br>• R4: Random forest.<br>• R5: Boosted trees.<br>• R6: Multivariate adaptive regression splines. | |
| **Performance measures:** | • Prediction accuracy scores via mean Brier scores with confidence interval,<br>and mean Nagelkerke R-squared with confidence interval.<br>• Model discrimination via the average loss in area under the<br>ROC-curve (AUC) with confidence interval.<br>• Model calibration via mean of calibration slopes (CS), calibration intercept,<br>integrated calibration index (ICI), and calibration in the large (CIL) all<br>also with confidence interval.<br>• Model sharpness via mean distance of the predicted probability to 1/2.<br>• Descriptive accuracy via mean true positive rate and mean false positive rate<br>with confidence intervals. | |

## 2.2 Aims

Our study aims to compare variable and feature selection methods across statistical and ML models concerning their performance for prediction. Of particular interest is the overall

performance of selection methods, with secondary aim the evaluation and comparison concerning descriptive scores.

## 2.3 Data generating mechanisms

The data will be generated in a three-step procedure. First, a statistical learning or ML model is estimated based on all data. Then, as a second step, a random subset of the observations in the STCS data is sampled, with predetermined sample size. Finally, the outcomes are generated according to the probabilities of the model estimated in the first step. The random subsample with predetermined sample size and generated outcome will then be used as a development dataset. The complement set to the random subsample is then used as evaluation dataset.

**2.3.1 Data generating mechanism for predictors.** The predictors are randomly sampled from the real dataset via subset sampling with sample size according to the corresponding planned sample size $n$ ($n \in \{250, 500, 1000\}$).

**2.3.2 Data generating mechanism for outcomes.** The DGMs are estimated based on all of the real data. The following six data-generating methods will be used.

- DGM1: Unpenalised logistic regression with continuous predictors using restricted cubic smoothing splines with three knots using the rms R-package [31].
- DGM2: Penalised logistic regression via LASSO (all continuous predictors linear). Parameter tuning of the $\lambda$ coefficient will be estimated via ten-fold cross-validation using the cv.glmnet function from the glmnet R-package [24].
- DGM3: Penalised logistic regression via RIDGE (all continuous predictors linear). Parameter tuning of the $\lambda$ coefficient will be estimated via ten-fold cross-validation using the cv.glmnet function from the glmnet R-package [24].
- DGM4: Random forest.
  The number of trees to grow is set to 500. Parameter tuning via the tuneRanger R-package [32] with 5-fold cross-validation for the parameters number of sampled candidate variables and minimum size of terminal nodes minimising the Brier score. The random forest itself is then constructed via the previous parameters with the randomForest R-package [33].
- DGM5: Boosted trees.
  We consider Friedmans stochastic gradient boosting machines using trees as the base learners, i.e. boosted trees, using the gbm R-package [34]. We will use sequences of 100 trees, and perform parameter tuning for two hyperparameters: interaction depth (determining the maximum depth of each tree) and shrinkage rate via cross-validation.
- DGM6: Multivariate adaptive regression splines.
  We plan to use the default parameters using generalized cross-validation using the original procedure outlined in Friedman's paper [35]. The degree of interaction is set to 1, pruning method backward pruning, maximum number of MARS terms at 200. The final model is then built using the earth R-package [36].

**2.3.3 Number of simulation runs $N$ calculations via Monte Carlo coverage.** To ensure the reliability of results in simulation studies, the number of simulation repetitions ($N = n_{sim}$) must be sufficiently large to achieve accurate estimates of performance measures. Specifically, the Monte Carlo error of these measures should remain within acceptable limits. We will use the calculation of the coverage of confidence intervals as a measure for this, which provides insight into the accuracy of interval estimation.

The Monte Carlo standard error(MCSE) for coverage can be estimated as [20,Table 6]:

$$\sqrt{\frac{\hat{\text{cover}}(1 - \hat{\text{cover}})}{n_{\text{sim}}}}, \tag{1}$$

where $\hat{\text{cover}}$ denotes the estimated coverage derived from the simulation. For $n_{\text{sim}} = 1500$ and $\hat{\text{cover}} = 95\%$, the MCSE is approximately 0.6%. In a worst-case scenario with $\hat{\text{cover}} = 50\%$, the MCSE is around 1.3%, which is still acceptable for simulation studies. Hence we plan to use $N = n_{sim} = 1500$ for all simulation settings (if feasible) after the protocol is published.

**2.3.4 Basis of the data: Study design, population and patient-related data.** This simulation study protocol is part of a nested project within the Swiss Transplant Cohort Study (STCS, www.stcs.ch). The STCS dataset encompasses prospectively collected information on all SOTs performed after 1st May 2008 [17,48]. All Swiss transplant centers, i.e., Basel, Bern, Geneva, St. Gallen, Lausanne and Zurich, contribute to data acquisition. The STCS was approved by the Ethic Committees of all participating institutions, i.e. Ethikkommission Nordwest- und Zentralschweiz EKNZ, Ethikkommission Bern, Ethikkommission Genf, Ethikkommission Ostschweiz EKOS, Ethikkommission Zurich. The responsible cantonal Ethics Committee (Ethikkommission Nordwest- und Zentralschweiz, Req. 2023-01812) approved this nested study on the 29th of September 2023. 3395 adult kidney-transplant recipients registered in the STCS between May 2008 and December 2021 providing written informed consent will be included in an anonymised form in the database that will be used for the simulation study. This data will be used for estimating the independent variable sampling distribution. Complete case analysis will be used for the estimations of the statistical learning/ML models.

## 2.4 Estimands and other targets

The main estimands of interest are (i) model prediction error, (ii) model discrimination and calibration, and (iii) descriptive correctness in terms of variables in the model.

## 2.5 Methods

In terms of methods, we first describe the planned variable and feature selection methods, and then the statistical learning/ML model frameworks

**2.5.1 Variable and feature selection methods.** In any variable selection we add age and sex by default as these variables are typically part of clinical prediction models [1,37]. We give a more detailed description of the variable selection strategies and methods to be used in the following.

- **Backward selection** starts with all candidate variables in the model, and variables are sequentially removed based on a rule, typically determined by a predefined threshold. This continues until only variables remain whose evaluation is above the threshold. The following scores will be used on logistic regression models:
  - **P-values**: Backward selection based on p-values is an often recommended method to be used with the thresholds of e.g. 0.2 or 0.5 [15,38]. The p-value computations will be based on a logistic regression based on methods from [39], using the rms R-package [31].
  - **AIC**: AIC is a variable selection score based on information theory that aims to balances model fit (via likelihood) and complexity by penalising the number of parameters [7].

- **Univariable threshold-based selection** evaluates a score for each predictor individually to determine its relationship with the outcome variable. Determined by a predefined threshold, the variables are put in the model or omitted. We will only use logistic regression-based univariate selection using P-values. The P-value from the logistic regression equals the P-value of a Pearson Chi-squared statistic for categorical variables [2].
- **k-best-based selection** evaluates a score for each predictor individually to determine its relationship with the outcome variable. The variables are ranked by a score, and the k best variables are put in the model with the rest omitted. Its simplicity and the control of the final number of variables used make it a convenient tool to construct filters for prediction models [1]. We decided the choices of $k = 7$ and $k = 14$ based on typical values of variables included for prediction models. The methods to rank the variables were based on subjectively conceived often used in the case of p-values, suitability in the case of the CAR score. For the others the choice is based on the applicability to both numerical and categorical variables as well as the performance of filters in [27]. We will use the following scores:
  - **CAR score**: CAR scores were proposed as a variable selection method for high dimensional models by means of correlation-adjusted marginal correlation estimation [40]. The corresponding computations will be obtained with the care R-package [40].
  - **Permutation importance**: Permutation importance is a score that is determined by evaluating the impact of randomly shuffling a feature's values on model performance. It was introduced by Breimann for random forests [5], and based on this a model-agnostic version was proposed [41]. A drop in performance after shuffling a feature's values indicates the feature's importance. We will use the mlr3 package in R [42] to compute it, using a version of the model agnostic permutation importance.
  - **Information gain**: Information gain score calculates mutual information, which ranks features by their ability to reduce uncertainty (entropy) in the target variable. Higher information gain indicates more informative features. We will use the mlr3 package in R [42] to compute it.
  - **Minimal joint mutual information maximisation (JMIM)**: The JMIM filter [43] was developed to potentially identify features with high relevance and low redundancy. It maximizes the mutual information between the selected features and the target variable while keeping the joint mutual information between the selected features as low as possible. We will use the mlr3 package in R [42] to compute it.
  - **Impurity importance**: Impurity importance is score originating from decision trees and random forests. It is used to rank features based on their ability to reduce node impurity, where impurity reflects the heterogeneity of class labels within a node in the tree construction, commonly measured by indices such as the Gini index. Features that contribute more significantly to reducing this impurity are deemed more important [5]. We will use the mlr3 package in R [42] to compute it.
  - **Symmetric uncertainty**: Symmetric uncertainty, an information-theory metric, normalizes mutual information to measure the dependency between variables. It ranges from 0 (no dependency) to 1 (full dependency) and is used in both classification and regression tasks [44]. We will use the mlr3 package in R [42] to compute it.
- **LASSO**: LASSO is a method that does both feature selection and regularization [24]. In the case of a linear model, it adds an L1 penalty for the coefficients the the residual sum of squares, whereas in our case of logistic regression it acts via penalized maximum likelihood. Thereby it can shrink some coefficients to zero, effectively selecting features while eliminating possibly irrelevant ones. The degree of regularization is controlled by a tuning

parameter, which determines how many features are retained in the model. Again parameter tuning of the $\lambda$ coefficient will be estimated via ten-fold cross-validation using the cv.glmnet function from the glmnet R-package [24].

- **Boruta**: Boruta is a feature selection algorithm based on Random Forests. It creates shadow features, i.e. copies of the original feature whose values were randomly shuffled, and compares their importance to the actual features. Features less important than the shadow features are iteratively removed, potentially ensuring that relevant features are retained [23].

**2.5.2 Statistical learning/ML models.** The statistical learning/ML models R1–R6 to be used are the same as the ones used to construct the data generating mechanism from Sect 2.3.2, except that in R1 the unpenalized logistic regression will consider all continuous predictors linear.

## 2.6 Performance measures

The scores can roughly be distinguished into prediction accuracy, model discrimination, calibration, sharpness, and descriptive accuracy. Many will be estimated using the val.prob function from the rms R-package [31]. The confidence intervals will be estimated via bootstrap.

Roughly, prediction accuracy measures how often a model correctly predicts outcomes, model discrimination evaluates how effectively the model distinguishes between groups, calibration measures whether the model's prediction frequencies align with actual outcomes, while descriptive accuracy assesses how well the model captures the true underlying distribution. Sharpness measures how concentrated probabilities are around the extreme values. A model making confident predictions ,e.g., probabilities near 90% or 10%, is sharper than one which frequently predicts probabilities near 50%, which reflects uncertainty [45].

The following performance measures will be used to estimate the prediction accuracy.

- Brier scores: The Brier score measures the accuracy of probabilistic predictions. For binary outcomes it corresponds to the mean squared error comparing the predicted probabilities with the actual outcome [2].
- Nagelkerke R-squared: Nagelkerkes R-squared measures the goodness of fit of a prediction model based on the likelihood [2]. The values are between zero and one, with values closer to one indicating that the predictions from the model fit the data better.

The following performance measure will be used to estimate model discrimination.

- AUC: The AUC is one of the most popular scores used for discrimination [46], and is also known as concordance or c-statistic. It can be interpreted as the probability that the predicted probability of a randomly selected "one" observation will exceed that of a randomly selected "zero" observation, correspondingly it ranges between zero and one with one a perfectly discriminating fit of the predictions [2].

The following performance measures will be used to estimate calibration.

- CIL: CIL is defined as the average predicted risk when compared with the overall event rate [47].
- Calibration slope and intercept: Calibration curves are often visually used to assess calibration. Calibration slope is an estimate of the distribution of the relative estimated risks, with a target value of one. The calibration intercept also asses CIL, with target zero.

- ICI: ICI is a calibration score that quantifies the average absolute difference between predicted probabilities and observed proportions [47].

The following measure will be used to estimate sharpness.

- Average sharpness is defined as the average distance to 1/2 measured in $L^1$ distance, i.e., when the observations are indexed by an index set $\mathcal{I}$ and $\hat{p}_i$ corresponds to the predicted probability for observation $i \in \mathcal{I}$, we define average sharpness as

$$Sharpness = \frac{1}{|\mathcal{I}|} \sum_{i \in \mathcal{I}} |2\hat{p}_i - 1| \tag{2}$$

The following performance measures will be used to estimate descriptive accuracy.

- True positive rate is the selection probability of the true predictors from the data-generating process.
- False positive rate is the selection probability of the false predictors from the data-generating process.

To enable the interpretation of results across simulation runs, we plan to organise summary statistics of the performance metrics across settings and visualise them, e.g., using heat plots.

## 3 Discussion

This protocol aims to examine the comparative performance of selected variable and feature selection methods for predicting patient outcomes, using six distinct data-generating processes. By aligning these processes with both regression and ML models, we ensure that the estimated models are not always fully nested within the data-generating models, thereby adding an important layer of complexity to our analysis. The variables considered in this study are complete, with no unobserved confounders by design. The data-generating mechanisms and simulated datasets, allow to evaluate the validity, model performance and generalization in various hypothetical scenarios.

Hence, overall, we will have 1500 simulations for each sample sizes of 250,500,1000 observations. In each we will have 6 data generating mechanisms (DGM1-DGM6), 27 variable selection strategies, for 6 regression/ML models (R1-R6), totalling in

$$1500 \cdot 3 \cdot 6 \cdot 23 \cdot 6 = 3726000 \tag{3}$$

simulation runs overall. Moreover, this approach provides flexibility to explore different data-generating processes and model-specific assumptions. It enables us to assess how well each method performs under potential model misspecification, a scenario that cross-validation and bootstrapping do not address as thoroughly. Additionally, simulating from a DGM allows us to control covariate distributions and relationships, offering more robust comparisons of model behavior across diverse conditions.

The datasets used for simulation are grounded in real data from STCS, and the variables are preselected based on input from clinical experts working in the field of infectious diseases. The final choice of variables was based on iterative consensus-building concerning clinical relevance and data availability. A complete list of the included variables, along with their clinical categories and rationale, is provided in Supporting information S2

However, this approach has two key limitations. First, the findings may not generalize to different contexts or fields beyond transplantation medicine. Second, while we aim to include a range of commonly used variable and feature selection methods, the techniques included are not exhaustive and may omit some other promising approaches.

Despite these limitations, the design of this study provides a robust framework for comparing variable and feature selection methods in viable, clinically relevant scenarios. The insights gained from this work will contribute to the optimization of predictive modeling in medicine, offering valuable guidance on the performance of these methods across different conditions.

## Supporting information

**S1: Visualisation of the simulation setup.** We describe the simulation setup in flowcharts. (PDF)

**S2: Description of population and variables on which the planned simulation is based.** We provide more information on the database and give information on dependent and independent variable with respect to generation of the DGMs 1–6 considered. (PDF)

## Acknowledgments

We thank Sandor Balog for a helpful exchange on feature selection in machine learning, and Korbinian Strimmer for a helpful exchange on the CAR score. We thank Georg Heinze, Willi Sauerbrei and Edwin Kipruto for helpful discussions on variable selection.

## Members of the Swiss Transplant Cohort Study

Patrizia Amico, Adrian Bachofner, Vanessa Banz, Sonja Beckmann, Guido Beldi, Christoph Berger, Ekaterine Berishvili, Annalisa Berzigotti, Françoise-Isabelle Binet, Pierre-Yves Bochud, Petra Borner, Sanda Branca, Anne Cairoli, Emmanuelle Catana, Yves Chalandon, Philippe Compagnon, Sabina De Geest, Sophie De Seigneux, Michael Dickenmann, Joëlle Lynn Dreifuss, Thomas Fehr, Sylvie Ferrari-Lacraz, Andreas Flammer, Jaromil Frossard, Déla Golshayan, Nicolas Goossens, Fadi Haidar, Jürg Halter, Christoph Hess, Sven Hillinger, Hans Hirsch, Patricia Hirt, Linard Hoessly, Günther Hofbauer, Uyen Huynh-Do, Franz Immer, Nina Khanna, Michael Koller, Angela Koutsokera, Andreas Kremer, Thorsten Krueger, Christian Kuhn, Arnaud L'Huillier , Bettina Laesser, Frédéric Lamoth, Roger Lehmann, Alexander Leichtle, Oriol Manuel, Hans-Peter Marti, Michele Martinelli, Valérie McLin, Katell Mellac, Aurélia Merçay, Karin Mettler, Sara Christina Meyer, Nicolas Müller*, Jelena Müller, Ulrike Müller-Arndt, Mirjam Nägeli, Dionysios Neofytos, Jakob Nilsson, Manuel Pascual, Rosmarie Pazeller, David Reineke, Juliane Rick, Fabian Rössler, Silvia Rothlin, Thomas Schachtner, Stefan Schaub, Dominik Schneidawind, Macé Schuurmans, Simon Schwab, Thierry Sengstag, Daniel Sidler, Federico Simonetta, Jürg Steiger, Guido Stirnimann, Ueli Stürzinger, Christian Van Delden, Jean-Pierre Venetz, Jean Villard, Julien Vionnet, Caroline Wehmeier, Markus Wilhelm, Patrick Yerly.

\* lead author: nicolas.mueller@usz.ch (NM)

## Author contributions

**Conceptualization:** Linard Hoessly, Jaromil Frossard, Simon Schwab.

**Formal analysis:** Linard Hoessly.

**Investigation:** Linard Hoessly, Jaromil Frossard, Simon Schwab, Frédérique Chammartin, Alexander Leichtle, Peter Werner Schreiber, Dionysios Neofytos, Michael Koller.

**Methodology:** Linard Hoessly, Jaromil Frossard, Frédérique Chammartin, Michael Koller.

**Project administration:** Linard Hoessly.

**Writing – original draft:** Linard Hoessly, Jaromil Frossard, Simon Schwab.

**Writing – review & editing:** Linard Hoessly, Jaromil Frossard, Simon Schwab, Frédérique Chammartin, Alexander Leichtle, Peter Werner Schreiber, Dionysios Neofytos, Michael Koller.

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
