## [Decision Letter · Decision Letter 0]

30 Jun 2025

PONE-D-25-23109Comparing variable and feature selection strategies for prediction - protocol of a simulation study in low-dimensional transplantation dataPLOS ONE

Dear Dr. Hoessly,

Thank you for submitting your manuscript to PLOS ONE. After careful consideration, we feel that it has merit but does not fully meet PLOS ONE’s publication criteria as it currently stands. Therefore, we invite you to submit a revised version of the manuscript that addresses the points raised during the review process.

We look forward to receiving your revised manuscript.

Kind regards,

Syed Nisar Hussain Bukhari

Academic Editor

PLOS ONE

Journal Requirements: 

 [This project (FUP 214) has been facilitated by the Swiss Transplant Cohort Study. The Swiss Transplant Cohort Study is supported by the Swiss National Science Foundation(SNSF grant 33CS30_201385, https://www.snf.ch), Unimedsuisse and the Transplant Centers.]. 

5. One of the noted authors is a group [the Swiss Transplant Cohort Study]. In addition to naming the author group, please list the individual authors and affiliations within this group in the acknowledgments section of your manuscript. Please also indicate clearly a lead author for this group along with a contact email address.

Reviewers' comments:

Reviewer's Responses to Questions

**Comments to the Author**

1. Does the manuscript provide a valid rationale for the proposed study, with clearly identified and justified research questions?

Reviewer #1: Yes

Reviewer #2: Partly

2. Is the protocol technically sound and planned in a manner that will lead to a meaningful outcome and allow testing the stated hypotheses?

Reviewer #1: Yes

Reviewer #2: Yes

3. Is the methodology feasible and described in sufficient detail to allow the work to be replicable?

Reviewer #1: Yes

Reviewer #2: Yes

4. Have the authors described where all data underlying the findings will be made available when the study is complete?

Reviewer #1: Yes

Reviewer #2: Yes

5. Is the manuscript presented in an intelligible fashion and written in standard English?

Reviewer #1: Yes

Reviewer #2: Yes

6. Review Comments to the Author

You may also provide optional suggestions and comments to authors that they might find helpful in planning their study.

Reviewer #1: This paper is acceptable because it presents a clearly structured and well-justified simulation study protocol. The use of the ADEMP framework ensures methodological rigor, the inclusion of both classical and ML methods proves its relevance. Also, the reliance on real-world transplantation data lends practical credibility. The study’s transparency, thorough documentation, and reproducibility also align well with the standards of a protocol paper.

Reviewer #2: I do not have any major concern. The protocol is precise and concise.

Three suggestions: 1) Provide a stronger rationale for the study. The current transformer-based ML methods have largely de-emphasizes the need for feature selection. Who would be most benefited from this study despite transformer-based, self-supervised models being available? 2) Please elaborate more on preselected variables based on opinions from clinical experts. 3) How will the results be presented for easy and clear comparison? Do the authors have a framework in mind?

7. PLOS authors have the option to publish the peer review history of their article (what does this mean?). If published, this will include your full peer review and any attached files.

Reviewer #1: **Yes: **Md Abdullah Akib

Reviewer #2: No

---

## [Author Response · Author response to Decision Letter 1]

2 Jul 2025

Response to Reviewers

Dear Editor,

Please find attached the revised manuscript "Comparing variable and feature selection strategies for prediction - protocol of a simulation study in low-dimensional transplantation data" which we submit for PLOS ONE.

We appreciate the comments and suggestions made by the reviewers and the academic editor(s). We have made corrections in the revised version in blue in the track-changed version. We give point-to-point replies for all issues raised in this rebuttal letter, ordered by main topic.

Sincerely,

Linard Hoessly, on behalf of all authors.

Journal Requirements:

1. Please ensure that your manuscript meets PLOS ONE's style requirements, including those for file naming. The PLOS ONE style templates can be found for main body and authors/affiliation.

A: We checked again the format requirement for the main body, added double spacing in latex, and adapted the equations environment for formulas. We checked again the format requirement for authors and affiliations and adapted the corresponding and the group authorship.

2. We note that the grant information you provided in the Funding Information and Financial Disclosure sections do not match. When you resubmit, please ensure that you provide the correct grant numbers for the awards you received for your study in the Funding Information section.

A: To comply with your request we added a section Funding Information after acknowledgements copying the financial disclosure.

\[This project (FUP 214) has been facilitated by the Swiss Transplant Cohort Study. The Swiss Transplant Cohort Study is supported by the Swiss National Science Foundation (SNSF grant 33CS30\_201385), Unimedsuisse and the Transplant Centers.]

A: We included a request to add the sentence "The funders had no role in study design, data collection and analysis, decision to publish, or preparation of the manuscript." in the cover letter.

A: We double-checked that the ethics statement is only in the Methods section.

5. One of the noted authors is a group \[the Swiss Transplant Cohort Study]. In addition to naming the author group, please list the individual authors and affiliations within this group in the acknowledgments section of your manuscript. Please also indicate clearly a lead author for this group along with a contact email address.

A: We double checked the author group, and provided lead author and contact email.

A: We checked the documentation on formatting references for PLOS ONE, checked for retractions, and removed all URL DOIs.

Reviewer #1:

This paper is acceptable because it presents a clearly structured and well-justified simulation study protocol. The use of the ADEMP framework ensures methodological rigor, the inclusion of both classical and ML methods proves its relevance. Also, the reliance on real-world transplantation data lends practical credibility. The study’s transparency, thorough documentation, and reproducibility also align well with the standards of a protocol paper.

A: Thank you, we appreciate your comments and critique.

Reviewer #2:

I do not have any major concern.

The protocol is precise and concise. Three suggestions:

1. Provide a stronger rationale for the study. The current transformer-based ML methods have largely de-emphasized the need for feature selection. Who would be most benefited from this study despite transformer-based, self-supervised models being available?

2. Please elaborate more on preselected variables based on opinions from clinical experts.

3. How will the results be presented for easy and clear comparison? Do the authors have a framework in mind?

A: Thank you, we appreciate your comments and critique.

Concerning (1), we added a sentence on line 13-16 in order to make the motivation more clear. Concerning (2) we added more details on preselection of variables on lines 356-360. For (3) we added a sentence on line 336-340 to give an idea of the framework we have in mind. Since the effectiveness of visualisations depends heavily on the underlying values and the simulations have not yet been run, it is currently not possible to be more specific on the final visual formats.

---

## [Editor Report · Decision Letter 1]

6 Jul 2025

Comparing variable and feature selection strategies for prediction - protocol of a simulation study in low-dimensional transplantation data

PONE-D-25-23109R1

Dear Dr. Hoessly,

We’re pleased to inform you that your manuscript has been judged scientifically suitable for publication and will be formally accepted for publication once it meets all outstanding technical requirements.

Kind regards,

Syed Nisar Hussain Bukhari

Academic Editor

PLOS ONE
---

## [Editor Report · Acceptance letter]

PONE-D-25-23109R1

PLOS ONE

Dear Dr. Hoessly,

I'm pleased to inform you that your manuscript has been deemed suitable for publication in PLOS ONE. Congratulations! Your manuscript is now being handed over to our production team.

Kind regards,

on behalf of

Dr. Syed Nisar Hussain Bukhari

Academic Editor

PLOS ONE